# Photobiomodulation Treatment in Chemotherapy-Induced Oral Mucositis in Young Haematological Patients—A Pilot Study

**DOI:** 10.3390/medicina58081023

**Published:** 2022-07-29

**Authors:** Paula Fiwek, Katarzyna Emerich, Ninela Irga-Jaworska, Dagmara Pomiecko

**Affiliations:** 1Department of Paediatric Dentistry, Medical University of Gdansk, 80-208 Gdansk, Poland; emerich@gumed.edu.pl (K.E.); dagmara.pomiecko@gumed.edu.pl (D.P.); 2Department of Paediatrics, Hematology and Oncology, Medical University of Gdansk, 80-208 Gdansk, Poland; ninela.irga-jaworska@gumed.edu.pl

**Keywords:** stomatitis, leukaemia, drug therapy, photobiomodulation

## Abstract

*Background and Objectives*: One of the most debilitating side effects of chemotherapy is oral mucositis (OM). Photobiomodulation (PBM) demonstrates high efficacy in the management of OM. The aim of the study was to investigate the incidence of oral mucositis and evaluation of the effectiveness of PBM therapy. *Materials and Methods*: A total of 23 children diagnosed with leukaemia or lymphoma affected by chemotherapy-induced OM were enrolled in the study. OM grade was assessed with the World Health Organization (WHO) scale. Patients completed an approved questionnaire, and blood cell counts were read every 2 days. OM lesions were treated with class IV laser therapy with a frequency of every 48 h and density of 2, 4, 8, 16 or 30 J/cm^2^. The level of pain was measured with VAS scale. *Results*: The 23 patients developed a total of 41 OM episodes with a mean duration of 7.61 days ± 4.70. Laser therapy showed a great reduction regarding pain and a better function of patients even with neutropenia. *Conclusions*: Oral mucositis represents a significant burden to children. PBM brings positive aspects for patients; however, the optimal treatment parameters require further study.

## 1. Introduction

Cancers are less frequent in children than in adults; about 1% of all cancer cases occur in children. Among all the childhood and adolescent cancers, haematological malignancies account for over 40% of diagnoses; leukaemia is responsible for about 29%, while lymphoma and reticuloendothelial neoplasms account for about 11% [1].

Although intensive chemotherapy (CT) regimens for children now have high efficacy, they are not devoid of side effects. One of the most debilitating for young patients and their caregivers is oral mucositis (OM). The prevalence of OM in children with cancer is between 52% and 80% [2]. This usually differs in the clinical picture. It can manifest as local irritation, erythema, and inflammation of the oral mucosa; however, it may also progress to generalized and painful erosions or ulcerations, resulting in hampered swallowing, chewing, and speaking [3]. 

Clinical symptoms usually appear within 7 days after chemotherapy begins [2]. Moreover, the chemotherapeutic regimen, age, the neutrophil count, and oral hygiene play an important role in the development and aggravation of mucositis. When OM occurs, the patients’ quality of life plummets, and they often require narcotic analgesia or fluid replacement. All this can cause multiple delays in chemotherapy and is frustrating for parents and doctors [4].

The approach to the topic of OM has changed over time. A new vision of the issue was presented by Sonis in 2009 [3]. He reported that mucositis was not a simple destruction of basal epithelial cells caused by chemotherapy and instead was a multistep process that took place in all tissues, especially non-keratinized tissue. This approach to OM provided a new perspective and an opportunity to prevent or manage the process at each step. 

However, the cause of mucositis still remains unidentified. Presently, many authors emphasize the correlation between gene polymorphisms and the toxicity of certain cytostatic drugs to the oral mucosa [5,6]. Most of the available studies to date have been performed on adult patients and might not represent the situation in young patients; however, unfavourable methotrexate (MTX) pharmacogenetics and a higher incidence of toxicity are well established in children with Down syndrome and acute lymphoblastic leukaemia [7].

Despite the frequency of OM, no standard therapeutic consensus has been determined. The current literature provides only a few evidence-based preventive and therapeutic methods for OM—in particular, maintaining proper oral hygiene, cryotherapy, keratinocyte growth factor supplementation, zinc supplementation, and low-level laser therapy [8,9]. Low-level laser therapy is also known as photobiomodulation (PBM) and applies to the use of red (660–700 nm) or near-infrared light (700–980 nm), usually produced by low-to-mid power lasers with a power density between 1 and 5 W/cm^2^, over the injury or lesions [10]. 

This therapy has demonstrated efficacy not only in the prevention of OM but also in the management of OM symptoms in children and adolescents with cancer who are receiving chemotherapy [11,12]. The Multinational Association of Supportive Care in Cancer and the International Society for Oral Oncology (MASCC/ISOO) and the European Society for Medical Oncology (ESMO, Lugano, Switzerland) recommend this method for prevention in patients treated with high doses of cytostatics, especially those preparing for bone marrow transplants [8,9]. Furthermore, the absence of reported side effects, as well as good tolerance by patients, appears to represent a strong advantage over other methods [13,14].

The aim of this study was to demonstrate the efficacy of PBM in the treatment of OM.

## 2. Materials and Methods

The study was approved by the Bioethical Committee of the Medical University of Gdansk (number NKBBN/364-32/2016). The patients enrolled in the study were hospitalized in the Paediatric Haematological Department at the Medical University of Gdansk in Poland during the period of one year (from February 2016 to February 2017). All the patients or their caregivers received and signed informed consent.

### 2.1. Design

This was a pilot study to evaluate the efficacy of photobiomodulation therapy on chemotherapy-induced oral mucositis in children. The treatment was provided using a class IV diode laser (LITEMEDICS -LAMBDA-Italy) with a continuous wavelength of 980 ± 10 nm (infrared) and 635 ± 10 nm wavelength for the aiming beam. The therapy was performed with a special photobiomodulation tip that was 1 cm in diameter with the use of a 300 μm Ø fibre and power indication from 0.1 to 1 W.

Furthermore, two protocols were established to assess the effectiveness of different dosages recommended by the laser’s manufacturer and presented in the literature [13] as follows:Protocol I: a fixed density of 30 J/cm^2^, power of 1 W, and time of 30 s.Protocol II: depending on the OM grade: grades 1, 2, 3, and 4 OM receiving a density of 2, 4, 8, and 16 J/cm^2^, respectively, with a power of 0.1 W and times of 20, 40, 80, and 160 s, respectively.

The dosage was assigned randomly and blinded in the patient group.

At first, the laser therapy was applied every 24 h; however, with the increasing number of patients, the frequency caused technical difficulties for the operator. Therefore, after six episodes of OM, it was changed to every 48 h.

### 2.2. Participants

The inclusion criteria for the study were aged between 3 and 18 years, diagnosed with leukaemia or lymphoma, receiving chemotherapy treatment, no photosensitizing drugs administered, OM developed to grade 1 on the World Health Organization (WHO) scale, a good general condition and cooperation, and parent’s/patient’s consent obtained.

No control group was formed for the study due to the lack of agreement from the Bioethical Committee and the fact that all the patients agreed to participate in the study after a detailed description of the PBM therapy had been provided.

### 2.3. Intervention

The PBM treatment started as soon as the haematologist asked for a consultation with the dentist or a patient reported a complaint regarding the oral mucosa. At each visit, a clinical investigation was performed, and the oral mucosa state and region were assessed using the WHO OM scale. In addition, a questionnaire was used to help to determine the grade of OM (Child International Mucositis Evaluation Scale—ChIMES), and the white blood cell count was reported [15]. 

The ChIMES was carried out with parents or caregivers, as well as cooperating children over 6 years of age, to whom the questionnaire’s questions were read by researchers. To investigate pain, a simple Visual Analogue Scale (VAS) was shown to the patient or, for children under the age of 5 years, to the caregiver. Pain was evaluated before each session of PBM. For objectivity, any analgesics received by a child were recorded.

As a safety measure during the laser sessions, the operator, the patients, the parents, and other people present in the work area were asked to wear protective glasses. During the PBM sessions, no contact with the inflamed mucosa was made, and the children were willing to help, often retracting the lip or cheek; however, considering the possibility of cross-transferring infections, the tip of the laser was properly secured. One OM episode was defined from the appearance of WHO grade 1 OM to a complete healing of all lesions (WHO grade 0) or until the patient had started a new chemotherapy session.

### 2.4. Research Questions

The study focused on the characteristics of the oral mucositis in children and the efficacy of PBM therapy. The aspects of PBM, such as the duration of the OM, the patient’s pain level, and the correlation between the neutrophil count and the healing of lesions, as well as the difference in the efficacy of the two treatment protocols, were taken into consideration.

### 2.5. Statistical Analysis

Statistical analyses were performed using Dell Statistica (a data analysis software system), version 13, (Dell Inc., StatSoft, Kraków, Poland, 2016). All the data are expressed as the means and standard deviations for the study participant variables, mucositis, and laser therapy sessions. The tests used to interpret the results concerning the duration of the OM, neutrophils, pain level, and protocols were as follows: one-sample test, correlation analysis, Spearman’s test, Wilcoxon test, Friedman ANOVA, and Mann–Whitney; statistical significance was considered at *p* < 0.05.

## 3. Results

### 3.1. Participants

A total of 23 patients met the inclusion criteria and were enrolled in the study. There were 14 girls (60.9%) and 9 boys (39.1%), with a mean age of 9.35 ± 4.86 years old. The vast majority of the examined patients were diagnosed with acute lymphoblastic leukaemia (19 patients, 82.6%), whereas lymphoma (three patients, 13%) and acute myeloid (one patient, 4.4%) were determined in the rest of the group. 

Due to the fact that some patients suffered from OM more than once during the study, 41 OM episodes in the group of 23 patients were assessed. Since each OM episode was different for a particular patient, the study results focus on the patients as well as on the episodes. All the patients tolerated the laser treatment well without any adverse effects or reactions.

### 3.2. General Results: All Patients

The cytostatic agents that appeared to be the most mucotoxic were, in order of frequency, MTX (21 OM episodes, 51.2%), cytosars with 6-mercaptopurine (10 OM episodes, 24.4%), and a combination of vincristine, daunorubicin, and L-asparaginase (10 OM episodes, 24.4%). Most of the OM episodes occurred throughout the consolidation phase (61%) of the treatment, while the rest—except for one—were observed during induction (37%) with only one (2%) during the maintenance phase.

The keratinized squamous epithelium was most frequently affected by OM, appearing on the buccal mucosa (58.6%) followed by the labial mucosa (50%). Other sites, such as the palate, tongue, gingiva, and oropharyngeal zone were also involved at a much lower frequency (Table 1).

The mean number of OM episodes per patient was 1.78 ± 0.90 (Table 2). The minimum number was one, and the maximum was four. The average number of PBM sessions per patient was 8.09 ± 5.75. However, it is worth noting the high skewness (As) of the results. The distribution for the number of sessions was strongly right-sided, since the vast majority of the patients had results below the mean for the group and had fewer than eight sessions; however, a few patients required multiple sessions—hence, the results. The number of sessions depended on the severity of the OM at the start of each episode.

The average number of PBM sessions required to complete the healing of the OM episode was 4.54 ± 2.27 (Table 2). The mean duration for an OM episode in the study was 7.61 days ± 4.70 starting from the appearance of the OM symptoms (≥grade 1 on the WHO scale) to complete healing of the lesions (grade 0 on the WHO scale). The results compared to a similar study group in the literature with a mean OM duration of 21 days, using a one-sample *t*-test, revealed a significantly shorter time of inflammatory symptoms than the reference values (t = −18.24, *p* < 0.05) (Table 3) [16].

Moreover, a dependence between the state of the mucous membrane and peripheral blood parameters was observed. When there was a suppressed white cell count, the first symptoms of inflammation of the oral mucosa began to appear. However, parents, as well as experienced physicians and nurses, reported faster wound healing and better function of patients associated with PBM sessions even in cases of stagnated neutropenia or a significant decrease in the neutrophil count occurring during the OM episodes as presented in Figure 1 and Figure 2.

The results of the correlation analysis (*p* < 0.05) between the number of sessions and the level of neutrophilia are presented in Figure 3.

There was no relationship between these two variables. Regardless of the number of sessions and the patient’s recovery, the level of neutrophils neither increased nor decreased; it remained at a relatively constant level. Furthermore, with each following visit, important pain relief was recorded, starting from the first session. The results of the VAS scale levels at the PBM sessions were analysed with Spearman’s correlation analysis (*p* < 0.05) and are presented in Table 4. With rho (186) = −0.50, *p* < 0.001, pointing towards a strong negative correlation, the results indicate that, the more the sessions in an OM episode, the lower the perceived pain.

In addition, the Friedman ANOVA test (*p* < 0.05) was used to demonstrate whether the laser treatment had an effect on pain relief with each consecutive session. The results, presented in Figure 4, indicate that the subsequent measurements on the VAS scale differed from each other with chi^2^ (N = 39, df = 2) = 55.13, *p* < 0.001, and that the pain intensity decreased with each laser session.

However, in order to demonstrate the comparison between each measurement, a Wilcoxon test was performed (*p* < 0.05) (Table 5). The results revealed, with *p* < 0.001 for each pair, that the second measurement was lower than the first but higher than the third, and the third measurement was lower than the first two.

### 3.3. Comparison of the Two Protocols

In the two regimes, the results were as follows:

Eighteen OM episodes were treated with Protocol I: two episodes of grade 1, six episodes of grade 2, eight episodes of grade 3, and two episodes of grade 4. Protocol II covered 23 episodes of OM, with three episodes of grade 1, thirteen episodes of grade 2, six episodes of grade 3, and one episode of grade 4. A comparison between the groups in terms of the number of sessions needed to heal the OM was performed using the Mann–Whitney test (*p* < 0.05); however, for a more thorough interpretation, the arithmetic means (M) are also shown in Table 6.

Regarding the significance score at *p* = 0.059 and the fact that it slightly exceeded the level of significance, a statistical tendency for the second protocol (2 J–16 J) to need fewer laser sessions to heal the OM appears to be accurate. Moreover, the two protocols were investigated taking into consideration the severity of the inflammation symptoms, regarding the area of the oral cavity and the intensity of the pain, with Spearman’s test (*p* < 0.05) (Table 7).

In the case of both protocols, the correlation was negative, revealing a statistically significant pain reduction with subsequent PBM sessions; however, in Protocol II (rho (89) = −0.71, *p* < 0.000), the patients had a much faster reduction in the pain level than those treated with Protocol I (rho (97) = −0.39, *p* < 0.000).

This corresponds with the results from Table 6, since the improvement was quicker, and the number of sessions required to complete OM healing was lower. Furthermore, with regard to the OM location, both protocols had a similar effectiveness in the buccal mucosa: Protocol I: rho (75) = −0.33, *p* < 0.004; and Protocol II: rho (57) = −0.45, *p* < 0.000. The case was the same for the lips: Protocol I: rho (75) = −0.25, *p* < 0.029; and Protocol II: rho (38) = −0.38, *p* < 0.018. The greater the number of sessions, the less severe the OM. Protocol I was more effective on the palate mucosa, with rho (8) = −0.73, *p* < 0.038, than on the tongue, which was more susceptible to Protocol II, presenting rho (12) = −0.73, *p* < 0.007; nonetheless, these locations had a rather low number of observations.

## 4. Discussion

As reported in the literature, the frequency of the occurrence of mucositis is varied [13]. According to Sonis et al., mucositis occurs in about 40% of patients treated with cytostatics due to cancer, while, according to other authors, it is estimated to occur in 30% to 75% of patients [17]. The prevalence of OM presented in this study was high, as it affected 67% (23 out of 34) of the children diagnosed with haematological malignancies who were hospitalized in the Paediatric Haematological Department of the Medical University of Gdansk during the 1 year period of the study. This frequency may result from the fact that almost one-third of the patients hospitalized during the time of study were diagnosed with leukaemia and undergoing severe cytostatic treatment. 

Most of the episodes that occurred were grade 2 or 3 on the WHO OM scale with painful erosions and ulcers. The higher the grade was, the longer the duration of the OM. Higher grades of OM predispose patients to a longer duration as they often influence the treatment and time of a patient’s hospital stay [18]. During grade 2 OM, patients frequently refused to maintain proper oral hygiene, and this was even worse at grade 3—although, according to Garrocho-Rangel et al., oral hygiene regimens significantly reduce the severity of mucositis [19]. 

In fact, in the group of patients who maintained better oral hygiene, OM proceeded with lower severity than in those who did not take care of their oral cavity. MTX appeared to be the most mucotoxic cytostatic agent; its high dosage and slow elimination rate are the two main factors that increase its toxicity. OM often appeared on the movable oral mucosa as expected according to the literature [20,21]. Furthermore, patients repeatedly started to feel burning of the oral mucosa or during swallowing before any sign of OM could be found in the mouth. 

The sensation was often associated with a decreasing white blood cell level and the beginning of inflammation of the mouth. The white cell count is likely not a cause of mucositis by itself; however, the two are connected because of the cycle of the cells. The membranes in the oral cavity and white cells have similar cell turnover. Therefore, if the patient received a mucotoxic agent when the decrease in white cells was observed, there was a greater risk of developing mucositis [21]. 

To precisely describe the state of the oral mucosa, a WHO scale was used in this study. This proved to be an objective and measurable scale as it created to capture the severity of OM, including the condition of the mucous membranes as well as the patient’s functionality. The WHO scale is broadly used in publications; in other cases, the National Cancer Institute’s (NCI) Common Toxicity Criteria Scale can also be found [2].

According to the literature, most of the existing remedies for OM, such as oral hygiene, rinses, analgesics, antibiotics, anti-inflammatory agents, and cryotherapy, are mostly supportive therapy for the problem [8,9]. Lately, the most promising remedy, human keratinocyte growth factor (KGF-1), has been shown to reduce the duration and severity of OM; however, according to Schubert et al., 63% of patients still develop mucositis, and the treatment is not devoid of side effects [22]. Moreover, KGF-1 therapy remains expensive and, therefore, is not affordable for many patients and departments. Consequently, there is still a strong need for other treatment methods. 

A favourable effect of laser therapy in chemotherapy-induced OM was first reported about 20 years ago [23]. Since then, the results of a number of trials have shown it to be efficacious [2,4,11,13]. Evaluation of the literature describing the clinical applications of laser therapy is complicated by the wide variations in methodology and dosimetry across different studies [24]. However, it is known to relieve pain, to have an anti-inflammatory effect, and to enhance wound healing due to its primary and secondary reactions in tissue [25]. 

The molecular and cellular mechanisms of PBM suggest that photons are absorbed by the mitochondria. They stimulate more ATP production and low levels of ROS, which then activate transcription factors, such as nuclear factor kappa-light-chain-enhancer of activated B cells (NF-κB), inducing many gene transcript products responsible for the beneficial effects of laser treatment. Nitric oxide is also involved in PBM and may be photoreleased in the respiratory chain [26]. 

Although the mechanism of laser therapy’s effects on a tissue has become clearer, many authors question its use in cancer patients, suggesting that it may increase cancer cell proliferation in vitro depending on the power output level of the laser and the number of applications [27]. However, most of the latest literature confirms that laser phototherapy is safe and effective in the treatment of chemotherapy-induced mucositis and, in fact, may increase the anticancer effect of the chemotherapy administered [28].

In that study, a class IV infrared laser with a continuous wave and an intraoral 1 cm^2^ tip was used to treat the OM lesions. The laser therapy was relatively inexpensive to use, excluding the initial purchase. The sessions were mostly performed at the patient’s bedside, as patients often were not able to stand or walk. The frequency of the PBM sessions was established in accordance with the references provided by the laser manufacturer and presented in the literature by Cauwels [13]. Other authors have used different frequencies, from twice a day to every 24 or 48 h [4,13]. The VAS used in the study was found to be very accurate—referring to the WHO scale—and proved to be sufficient for determining pain. 

Importantly, it was also understandable and clear for the patients and their caregivers as described in the literature [29]. ChIMES was helpful in determining the exact cause of pain, as a sore throat was frequently mistaken for OM symptoms. PBM was shown to have a good analgesic effect in the presented group of patients. With each subsequent laser session, patients reported a significant reduction in the pain level. In addition, patients noticed faster wound healing compared to previous cycles without the use of PBM. However, this effect of the laser treatment can only be compared to the mean OM duration reported in the literature; one limitation of the study was the lack of a control group. 

However, while in the literature, chemotherapy-induced mucositis was described to last 21 days after therapy, the mean duration of OM presented in the patients in the study was 7.61 days ± 4.70 [19,30]. In addition, the improvement in the condition of the oral mucosa was not due to an increase in the number of neutrophils, which remained relatively constant. As described by Pels, wounds and ulcers that are difficult to heal are related to the blood parameters—in particular, lesions of the mucositis type are dependent on the level of neutropenia [30].

At the same time, a clinical difference in pain reduction and the sessions required to complete healing of the OM in favour of Protocol II (2 J–16 J/cm^2^) was observed. Moreover, while both protocols appeared to be equally effective on buccal and lip mucosa, there seemed to be a difference between two oral cavity locations, with Protocol I being more advisable on the palate and Protocol II being more advisable on the tongue mucosa. It should be emphasized that the confirmation of such a dependence requires further research in a larger group of patients.

## 5. Conclusions

While these results are encouraging, further study is needed to truly establish the efficacy of this mucositis treatment on a more homogenous group of patients. However, a certain difficulty should be taken into consideration, which is that the time required to perform one PBM session per patient in the case of diffuse OM changes was up to 20–30 min with pauses. The latest systematic reviews indicate facilitation of the delivery of laser treatment by extraoral application and a larger laser tip, which would make the treatment—and especially the prevention—of OM much easier [26].

Laser therapy is a breakthrough in the management of OM; however, it is also a well-known tool in medicine and dentistry. This study showed the positive aspects of laser therapy for patients and described the difficulties and limitations for the doctors who perform it. There is a strong need for more research to determine the effects of modifying the exact laser parameters on the effectiveness of PBM to treat and prevent OM.

## Figures and Tables

**Figure 1 medicina-58-01023-f001:**
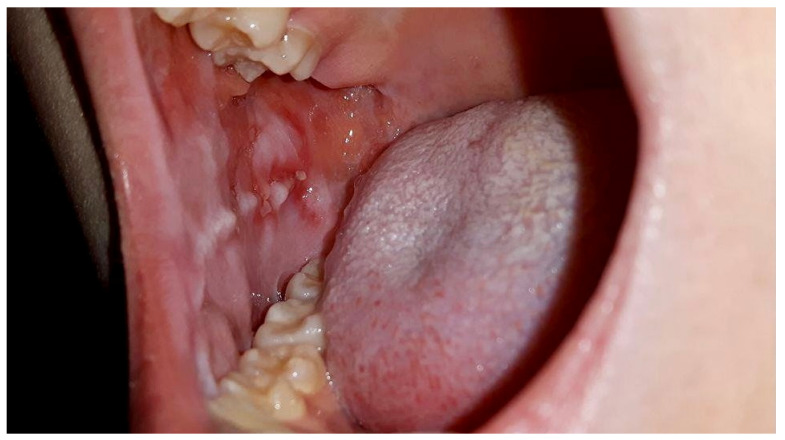
Intraoral photograph showing buccal mucosa with oral mucositis grade 3 (WHO scale) in a 15 year old girl, one week after methotrexate therapy, before PBM sessions, with a neutrophil count of 0.36 × 10^9^/L. The patient was not able to eat solid foods, only liquids, with a pain level at VAS = 8.

**Figure 2 medicina-58-01023-f002:**
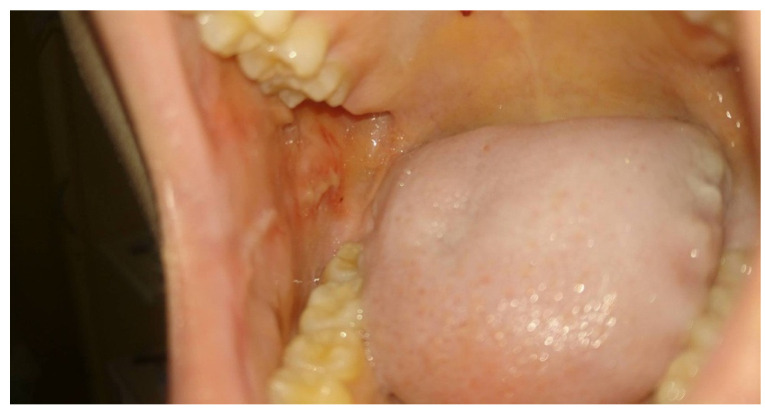
The same patient after two PBM sessions (96 h later) with OM grade 1 (WHO scale), a neutrophil level of 0.43 × 10^9^/L, and pain at VAS = 3, eating solid food.

**Figure 3 medicina-58-01023-f003:**
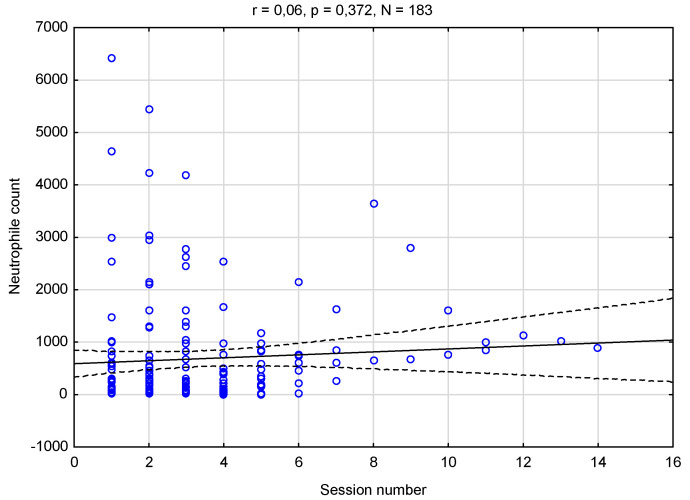
Correlation between the number of PBM sessions and the level of neutrophils; correlation analysis (*p* < 0.05).

**Figure 4 medicina-58-01023-f004:**
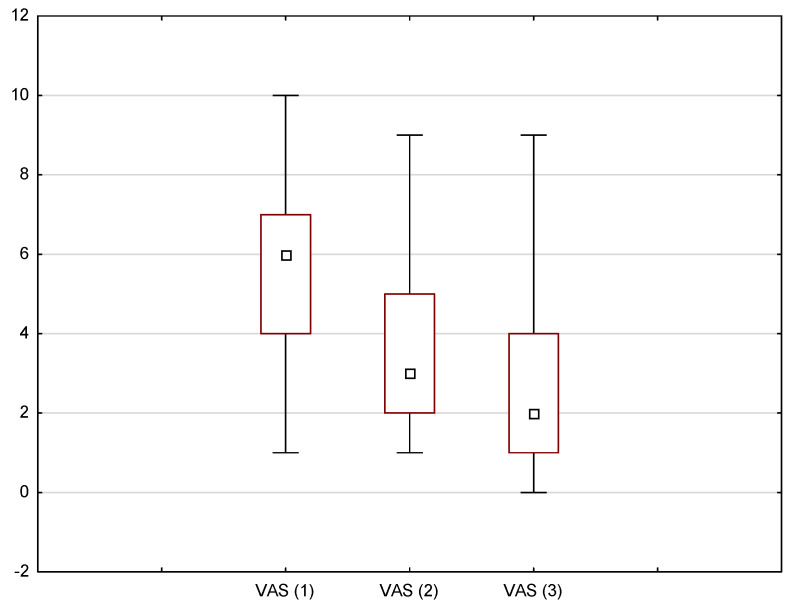
The level of pain intensity (measured with the VAS scale) after subsequent laser sessions; Friedman ANOVA test, *p* < 0.05.

**Table 1 medicina-58-01023-t001:** The frequency and severity of OM in different locations of oral mucosa.

	Location of the OM Lesions											
WHO Scale	B		L		P		T		G		O	
	*N*	%	*N*	%	*N*	%	*N*	%	*N*	%	*N*	%
0	70	37.63	90	48.39	156	83.87	161	86.56	183	98.39	172	92.47
1	44	23.66	42	22.58	17	9.14	11	5.91	1	0.54	14	7.53
2	36	19.35	35	18.82	8	4.30	13	6.99	2	1.08		
3	29	15.59	16	8.60	4	2.15	1	0.54				
4	7	3.76	3	1.61	1	0.54						
Total with WHO > 0	109	58.60	93	50.00	29	15.59	25	13.44	3	1.61	14	7.53

B—buccal mucosa; L—lips; P—palate; T—tongue; G—gingiva; and O—oropharyngeal area.

**Table 2 medicina-58-01023-t002:** Statistical description of OM and PBM relations in the group of all patients and episodes.

Variable	N	M	Me	Min	Max	SD	As	K
OM episodes/patient	23	1.78	2.00	1.00	4.00	0.90	0.88	−0.11
PBM sessions/patient	23	8.09	6.00	3.00	29.00	5.75	2.41	7.41
PBM sessions/episode	41	4.54	4.00	2.00	14.00	2.27	2.42	7.73
OM duration/episode	41	7.61	6.00	3.00	28.00	4.70	2.90	9.85

N—number of observations; M—average; Me—median; Min—minimum value; Max—maximum value; SD—standard deviation; As—skewness; and K—kurtosis.

**Table 3 medicina-58-01023-t003:** Comparison of the average number of days needed to heal an episode of OM in this study with the reference values in the literature (one-sample *t*-test).

Variable	M	SD	N	SE	Reference Value	t	df	*p*
Duration of OM (days)	7.61	4.70	41	0.73	21.00	−18.24	40	0.000

M—average; SD—standard deviation; N—number of observations; SE—standard error; t—test; df—degrees of freedom; and *p*—level of statistical significance.

**Table 4 medicina-58-01023-t004:** Correlation between the number of sessions during an episode and the level of pain experienced (VAS); Spearman’s correlation analysis (*p* < 0.05).

	Stat.	Number of Sessions in the Episode				
		All Patients	M	F	Up to 9 y	10 + y
VAS	rho	−0.50	−0.33	−0.65	−0.69	−0.43
	*p*	0.000	0.003	0.000	0.000	0.000
	N	186	81	105	57	129

rho—Spearman’s test value; *p*—level of statistical significance; N—number of observations; M—male; and F—female.

**Table 5 medicina-58-01023-t005:** Comparisons between consecutive pain intensity on the VAS scale; Wilcoxon test, *p* < 0.05.

	VAS (1) R = 2.81	VAS (2) R = 2.00	VAS (3) R = 1.19
VAS (1)		<0.001	<0.001
VAS (2)	<0.001		<0.001
VAS (3)	<0.001	<0.001	

*p*-values for the Wilcoxon test; R—average rank; and VAS—visual analogue scale.

**Table 6 medicina-58-01023-t006:** Comparison of the OM episodes treated with Protocols I and II in terms of the number of sessions needed to complete healing of OM; Mann–Whitney test (*p* < 0.05).

Variable	Protocol I			Protocol II			Statistics	
	N	R	M	N	R	M	Z	*p*
Number of sessions/OM	18	25.03	5.39	23	17.85	3.87	1.89	0.059

N—number of observations; R—average rank; M—average; Z—value of the Mann–Whitney test; *p*—level of statistical significance; Protocol I—30 J/cm^2^; and Protocol II—2, 4, 8, and 16 J/cm^2^.

**Table 7 medicina-58-01023-t007:** Dependencies between the number of sessions during an episode and the level of severity of inflammatory symptoms in different locations of the oral cavity and the severity of pain; Spearman’s test (*p* < 0.05).

	Stat.	Severity of OM in Different Locations and the Intensity of Pain				
		B	L	P	T	VAS
Number of sessions (Protocol I)	rho	−0.33	−0.25	−0.73	−0.39	−0.39
	*p*	0.004	0.029	0.038	0.098	0.000
	N	75	75	8	19	97
Number of sessions (Protocol II)	rho	−0.45	−0.38	−0.02	−0.73	−0.71
	*p*	0.000	0.018	0.932	0.007	0.000
	N	57	38	25	12	89

rho—Spearman’s test value; *p*—level of statistical significance; N—number of observations; B—buccal mucosa; L—lips; P—palate; T—tongue; and VAS—visual analogue scale.

## Data Availability

All the data generated or analysed during this study are included in this published article.

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
