# Peer review of "Photobiomodulation Treatment in Chemotherapy-Induced Oral Mucositis in Young Haematological Patients—A Pilot Study"

_medicina, 2022, doi:10.3390/medicina58081023_

Round 1

Reviewer 1 Report

There is not scientific research to support the use of 4, 6, 16 e 30 J/cm2, the 980nm and 0.1 to 1 W.

Reviewer 2 Report

Dear Authors the paper has been improved as suggested.

Thank you for your paper

This manuscript is a resubmission of an earlier submission. The following is a list of the peer review reports and author responses from that submission.

Round 1

Reviewer 1 Report

Dear Authors

  • Today it is agreed on to call Low level laser therapy "photobiomodulation. Can you please correct this in your Manuscript.
  • The place of the pictures is not adequate. Please check the guidelines of the MDPI. 
  • The authors names and affiliations section is missing.
  • Overall, the template do not follow the MDPI guidelines.
  • Remove sentence 61 "most recent and promising one." This is not very scientific.
  • Line 113: remove "LiteMedics" and please write in details the brand name of the company.
  • The energy density was not mentioned in the parameters. The time was not mentioned and the formula is not very accurate. What you need to choose is the energy density that the tissue will receive and everything else will be made accordingly.
  • There is no statistical analysis ?
  • The results are not represented accurately.
  •  

The scientific content of the article is interesting. However, how it is written and the data analysis needs to be seriously corrected.

Reviewer 2 Report

Medicina-1675112

Title: Chemotherapy-induced oral mucositis (OM) in young 3 hematological patients – prevention and treatment 4 with low-level laser therapy (LLLT).

The paper is directed to the Medicina. The paper is proposing to investigate the incidence of oral mucositis and evaluation of the effectiveness of low-level laser therapy (LLLT) on oral mucositis. The authors concluded that oral mucositis represented a significant burden to children. LLLT brings positive aspects for patients; however, optimal treatment parameters require further study.

In fact, the oral mucositis and the photobiomodulation therapy (PBMT) are aspects important very much to the oncologic patients.

The paper had a good view, but there is concepts not applied and is not well structured.

In tittle, is described the PBTM related to the prevention and treatment of oral mucositis, but, prevention is not addressed in the text;

Figures is not frequently presented in introduction;

There is not scientific research to support the use of 4, 6, 16 e 30 J/cm2, the 980nm and 0.1 to 1 W.

There are various cytostatic agents: MTX, cytosars with 6-mercaptopurine, and a combination of vincristine, 147 daunorubicin and L-asparaginase and this represent a bias.

Reviewer 3 Report

Dear Authors

thank you for your paper. I've found it very interesting, well organized and well described. The topic is important although laser use in oral mucositis is well described in litarature. Your paper add something to the current literature, but I suggest to expand laser description from a general point of view especially regarding further use in the oral cavity, and also to add more picture (possibly of the cheek mucosa) as well some control after therapy.Thank you again for your paper